# The experiences of hospital staff with decision-making concerning patient enrolment in hospital at home services: A complex and dynamic process

Lillian Karlsen[1,2]*, Bente Prytz Mjølstad[3], Bjarte Bye Løfaldli[2], Anne-Sofie Helvik[1,3]

1 Department of Public Health and Nursing, Faculty of Medicine and Health Sciences, Norwegian University of Science and Technology, Trondheim, Norway, 2 The Centre for Health Innovation, Kristiansund, Norway, 3 General Practice Research Unit, Department of Public Health and Nursing, Norwegian University of Science and Technology, Trondheim, Norway

* Lillian.karlsen@helseinnovasjonssenteret.no

## Abstract

### Background

Hospital at home care services offer a potential solution to the problem of strain on hospital beds while simultaneously enhancing patient outcomes. Nevertheless, implementation of the hospital at home care model is associated with several challenges. One such barrier involves patient enrolment, particularly during the initial stage of service operation. Due to their frontline experience, healthcare professionals possess valuable insights that can help us understand and address this challenge. This study aimed to explore the experiences of hospital staff in the decision-making process concerning patient enrolment in hospital at home.

### Methods

In total, 22 semi-structured individual interviews were conducted with hospital staff members between January and May 2022 at the participants' workplace or in a public office depending on their preferences. Data were analysed using reflexive thematic analysis.

### Results

We identified four themes pertaining to the experiences of hospital staff with the decision-making process concerning patient enrolment in hospital at home: "beneficial for the patients; an important motivating factor", "patient eligibility; prioritizing safety", "contextual factors within hospital ward units; opportunities and limitations", and "collaboration with municipalities; crucial but challenging".

### Conclusions

Hospital staff described a complex and dynamic decision-making process when considering patient eligibility for enrolment to hospital at home services. The findings highlight both

**Data Availability Statement:** Data cannot be shared publicly due to Norwegian law regarding confidentiality and privacy of participants. Interview data cannot be shared publicly as participants have not given consent for their transcripts to be shared

in a public repository. Data (in Norwegian language) are available from the Department of Public Health and Nursing at the Norwegian University of Science and Technology (ism-post@mh.ntnu.no) or the corresponding researcher (lillian. karlsen@helseinnovasjonssenteret.no) for researchers who meet the criteria for access to confidential data. For more information about data requests relating to this study, the Norwegian Agency for Shared Services in Education and Research (SIKT) can be contacted at https://sikt. no/en/om-sikt/contact-us#Contactform.

**Funding:** This study was supported by the Research Council of Norway, grant number 327215 (https://www.forskningsradet.no/). Additionally, support was provided by The Centre for Health Innovation (Helseinnovasjonssenteret). LK was the recipient of these grants. The funders had no role in study design, data collection and analysis, decision to publish, or preparation of the manuscript.

**Competing interests:** The authors have declared that no competing interests exist.

barriers and enablers pertaining to this process and emphasize the need to provide support to hospital staff as they navigate this complex situation. A key finding pertains to the critical importance of high-quality decision-making in ensuring positive outcomes and the overall effectiveness of hospital at home care services. Additionally, this study proposes a deeper exploration of the ethical considerations associated with balancing the goal of patient safety with that of equitable access to high-quality, person-centred care within the context of hospital at home.

## Introduction

In response to the global challenge of acute hospital beds shortages and limited hospital capacity, there is a concerted effort to explore alternative healthcare models aimed at reducing hospital stays [1].

One such proposed solution is the Hospital at Home (HaH) care model, also referred to as Hospital in the Home [2], Home Hospital [3], Home Hospitalization [4], and Hospital-based Home Care [5]. HaH offers an alternative care setting for individuals who would otherwise require hospitalization, providing acute hospital-level care at home for patients stable enough to be at home [3, 6, 7]. This helps alleviate the demand for hospital beds.

HaH is a complex intervention with multiple interacting elements within the healthcare system [8]. Acute care in a patient's home environment has been found to yield positive outcomes, including reduced length of hospital stays [9], and promotion of person-centred care [10]. Additionally, it has favourable impacts on readmission rates, quality of care, patient safety, and patient satisfaction [11–13].

Approaches to HaH vary widely both within and across countries and healthcare settings, including variations between high-income versus low-to middle-income countries, single-payer versus multiple-payer systems, and urban versus regional or rural contexts [14]. Differences also exist in referral mechanisms, boundaries with other services, and definitions of HaH [14]. Under the HaH umbrella, programs differ in the illnesses and patient groups they address, the interventions they provide, and the stakeholders involved in care delivery [15]. Additionally, the duration of treatment, the technology involved, organizational structures, and funding mechanisms vary [16].

In Norway, 17.5% of the population is aged 65 or older, and the country has the highest life expectancy in Europe [17]. With acute hospital bed availability below the EU average and occupancy rates at 80% in 2020 [17], there is a growing need for alternatives to in-hospital treatment [18, 19]. While HaH care pilot programmes exploring various approaches are ongoing, nationwide implementation is pending [20]. One innovative approach, central to this study, enables patients to receive intravenous antibiotic treatment at home, partially substituting for hospital care. This service integrates into the existing healthcare system, relies on collaborative partnerships between hospital and municipalities, and involves patients and family caregivers in the care process [21].

Despite the potential benefits of the HaH care model, its implementation faces challenges, limiting patient inclusion and adoption in several countries [22]. Successful implementation of complex healthcare interventions relies heavily on healthcare professionals' attitudes and perceptions [23]. Therefore, we should obtain a comprehensive understanding of their experiences in the HaH context.

During the initial phase of HaH service operation, healthcare professionals face challenges pertaining to patient enrolment [24]. These challenges are partly linked to physicians' concerns and hesitancy in determining the most suitable care model, which can influence the effectiveness of HaH [25, 26]. Clinical decision-making¸ an interprofessional process [27], plays a pivotal role in selecting the best treatment [28]. To fully understand the decision-making process, it is crucial to explore the steps leading up to the decision [29]. This pre-decisional phase involves gathering adequate information, exploring preferences, and active discussions among various stakeholders [29].

Our knowledge of the challenges faced by healthcare professionals during patient eligibility considerations for HaH enrolment remains limited. Thus, this study focuses on gaining insight into the experiences of hospital staff who are involved in this critical work, with the goal of obtaining a deeper understanding of the barriers and enablers they encounter. This research carries implications for enhancing HaH adoption and integration, aligning with the broader goal of healthcare delivery improvement in both Norwegian and global contexts.

The aim of this study is to explore hospital staff members' experiences in the decision-making process concerning patient enrolment in a novel HaH care program.

## Materials and methods

### Design and setting

A qualitative, explorative interview study was conducted from a hermeneutic phenomenological perspective. The aim was to understand the meaning of the lived experiences of hospital staff involved in decision-making regarding patient enrolment in HaH. Hermeneutic phenomenology goes beyond obtaining a descriptive understanding of a phenomenon by delving into interpretation, exploring individuals 'consciousness, and understanding how they confer meaning to aspects of their lives. This approach allows for an exploration of personal perceptions regarding the phenomenon under investigation [30]. To improve the trustworthiness of the findings, the Consolidation Criteria for Reporting Research (COREQ) guidelines were adhered to during the reporting process [31] ("S1 Checklist").

Norway's healthcare system, rooted in the Nordic welfare model and primarily funded by public sources [32], operates with responsibilities divided between national and local governments. Regional health authorities manage specialized healthcare, while municipalities oversee primary health care, including social services and home care [33].

The HaH care program central for this study was developed at a region in Mid-Norway and implemented from 2018 and 2022, using a co-creative service design approach. This approach involved engaging various stakeholders throughout the process of designing, developing, and testing this novel solution [34]. The program is "treatment-oriented" [35], providing long-term intravenous antibiotic therapy for a range of infectious diseases to adult patients admitted to either an internal medicine ward or a post-orthopaedic surgical ward. The treatment utilizes peripherally inserted central catheters and ambulatory pumps, facilitating intermittent dosing, keeping the vein open, and allowing patients and their family members to actively participate in the treatment process [36]. Healthcare professionals involved in decision-making concerning patient enrolment in HaH also bear responsibilities such as preparing, coordinating, and monitoring care until the end of treatment. Delivery of treatment and care in the HaH care program involves collaborative partnership with a hospital pharmacy, municipal home nurses, and a municipal response centre. Patients are equipped with a digital safety alarm for emergency management, but no remote patient monitoring tools are included. Throughout all phases of HaH treatment, the hospital retains full clinical responsibility.

Eligibility criteria for patient enrolment in the HaH care program is detailed in Table 1.

**Table 1. Patient eligibility criteria for HaH treatment.**

| |
| --- |
| Over 18 years of age with the capacity to consent |
| Medical, post-surgical, or neurological diagnoses |
| In need of prolonged intravenous antibiotic treatment |
| In a stable condition |
| Able to adhere to the care plan |
| Motivated and willing |
| Suitable home environment |

## Sample and data collection

Hospital staff who worked at the hospital that provide the HaH care program, possessing relevant experience with HaH care, were purposively selected to ensure diversity in age, gender, and professional experience. These staff members were invited to participate, face to face or by telephone, by people in the administration level at the hospital independent of the research team. A total of 22 hospital staff members were recruited from January 28,2022 and May 1, 2022, including 9 physicians, 11 nurses, and 2 participants representing other professional disciplines (Table 2).

Data were collected through semi-structured individual interviews, using an interview guide developed by the authors specifically for this study, drawing on previous research, clinical experience, and the specific aim of this study. The guide and interviews were developed for this study and has not previously been published elsewhere. Peers reviewed the questions and provided feedback. A pilot test of the guide was also conducted before the interviews. The main questions are detailed in Table 3, although a flexible approach was used, allowing for follow-up questions to elaborate on previous answers [37].

**Table 2. Characteristics of the study participants (n = 22).**

| Hospital staff | |
| --- | --- |
| **Health care profession** | |
| Nurse | 11 |
| Physician | 9 |
| Other | 2 |
| **Length of current work position *** | |
| 1–10 | 16 |
| 11–20 | 4 |
| 21–30 | 2 |
| **Age *** | |
| 20–29 | 3 |
| 30–39 | 7 |
| 40–49 | 5 |
| 50–59 | 4 |
| 60–69 | 3 |
| **Gender** | |
| Female | 18 |
| Male | 4 |

* In years

**Table 3. Thematic interview guide.**

| | |
|---|---|
| 1. | What role and responsibility do you have at the hospital and in patient enrolment in HaH? |
| 2. | How do you experience the work of evaluating patient eligibility for HaH treatment? |
| 3. | How do you experience collaboration among professionals and partners when evaluating patient eligibility for enrolment in HaH? |

Interviews were conducted face-to-face at participants' workplaces or in public offices, depending on their preferences. Following each interview, notes were made to capture any thoughts or impressions that could have impacted the quality of the dialogue, along with initial interpretations of the key elements communicated by participants.

The sample size was guided by an assessment of information power [38]. The first author (LK) conducted and audio-recorded all interviews, with subsequent verbatim transcription. A few interviews were transcribed externally. The interviews lasted from 21 to 81 minutes, with a median duration of 50 minutes. No repeat interviews were conducted.

## Analysis

Data were analysed using reflexive thematic analysis, allowing for the creation of themes, which are patterns of shared meaning united by a central concept [39]. In the process of data engagement, coding, and theme development, the authors iterated across six phases as described by Braun and Clarke [40].

Initially, the text was thoroughly read several times to enable the researchers to immerse themselves in and familiarize themselves with the data. Then, each transcript was manually coded, marking segments of the text that seemed to be relevant to the research objective. Both semantic and latent codes were identified through two rounds of coding. Codes across all the transcriptions were clustered into initial candidate themes, which were further abstracted, refined, and labelled as main themes.

The first author (LK) was mainly responsible for the analysis, who collaborated with all authors (ASH, BPM, and BBL) during the entire process of analysis. All authors met to discuss the preliminary themes until all agreed on the final themes. Table 4 presents an example of the analytical process.

**Table 4. Example of the analytical process.**

| Transcript | Code | Initial theme | Theme |
|---|---|---|---|
| *"It is purely positive for patients to get out of a hospital if they absolutely do not have to be hospitalized then. All the patients I have met say that they eat much better, they are much more mobile, much more active and they have their family around them (...)"* | Patient sleep better<br>Patients eat better<br>Patients are more mobile<br>Patients are more socially active | Benefits of HaH for patients | Beneficial for the patients; an importing motivating factor |
| *"You can't really mix the hospital ward into that (the decision-making process for patient enrolment); it is like a refugee camp sometimes. With ten patients too many and nurses running in the corridors. The physicians are waiting and will soon go in and operate... so it's like that... everything must be done in passing, I feel. No one sits down and focuses on this task. They are always on their way... somewhere... right?"* | Difficult to integrate<br>A high pace in the ward | Reflections on organizing the HaH service | Contextual factors within hospital ward units; opportunities and limitations |
| *"After all, patients are generally discharged much earlier than they were a few years ago. So, I understand the municipalities if they feel that they are given tasks for which they are not paid..."* | Transfer of work from hospital to municipalities | Collaboration with and attitudes towards the municipalities | Collaboration with hospitalities; crucial but challenging |

### Ethics and reflexivity

This study was registered and conducted in accordance with the protocol of the Norwegian Centre for Research Data (ref. no. 183099). The Regional Committee for Medical Research Ethics Central Norway determined that the study did not require ethical approval as it was consequently not covered by the Health Research Act, paragraphs 2 and 4 (ref. no. 267185). All participants received written and verbal information regarding the purpose of the study, their right to decline participation, the confidentiality of the data, and their right to withdraw at any time. Prior to the interviews, written informed consent was obtained from each participant. Data were anonymized by using identification codes to ensure confidentiality.

Our pre-understanding as a research team was shaped by our diverse professional background and scientific expertise. Comprising a cancer nurse, a nurse specialized in geriatric care, a general practitioner, and a neurobiologist focusing on healthcare innovation and implementation, our team brought varied perspectives to the study. Additionally, three authors, ASH, BPM, and BBL, are senior researchers, with ASH and BPM having extensively experience in qualitative research. Both the first author (LK) and author BBL had prior knowledge of the development and implementation process of the HaH program in Mid-Norway.

## Results

We identified four themes pertaining to hospital staff's experiences with the decision-making process concerning patient enrolment in HaH: "beneficial for the patients; an important motivating factor", "patient eligibility; prioritizing safety", "contextual factors within hospital ward units; opportunities and limitations", and "collaboration with municipalities; crucial but challenging", where the first theme appears to serve as a foundational element that supports the three subsequent themes (Fig 1).

### Beneficial for the patients; an important motivating factor

The general attitude among hospital staff suggested that HaH was positive for patients based on their experiences with the disadvantages of hospitalization. These disadvantages encompassed emotional toll of separation from family members, the risks of hospital-acquired infections, delirium, and functional decline, especially for patients who experienced extended stays, often in isolation: "Lying for two months to receive antibiotic treatment is not good for anyone. We experience that patients undergo psychological strain all the time. Patients get depressed by lying here" [18].

Hospital staff observed that patients receiving HaH treatment provided positive feedback and positive clinical outcomes. A well-experienced physician remarked, from a medical perspective, "(. . .) hospital at home is not inferior; quite the opposite (. . .)" [15]. Furthermore, the familiar environment of home was perceived as facilitating health promotion. Being treated at home improved patients' appetites and sleep and increased their physical activity, and enhanced social engagement, all associated with faster recovery.

Additionally, the home context allowed patients to resume their daily routines, which were viewed as reestablishing a sense of normalcy and strengthening their emotions of a meaningful life despite the health-related challenges. This restoration of normalcy enabled patients to shift their focus from illness and limitations to health, wellbeing, and personal resources:

> "Patients can live their lives. So, the disease or the treatment becomes a small thing instead of taking over their everyday lives. You can be healthy even if you are a little sick and have a life outside the hospital" [14].

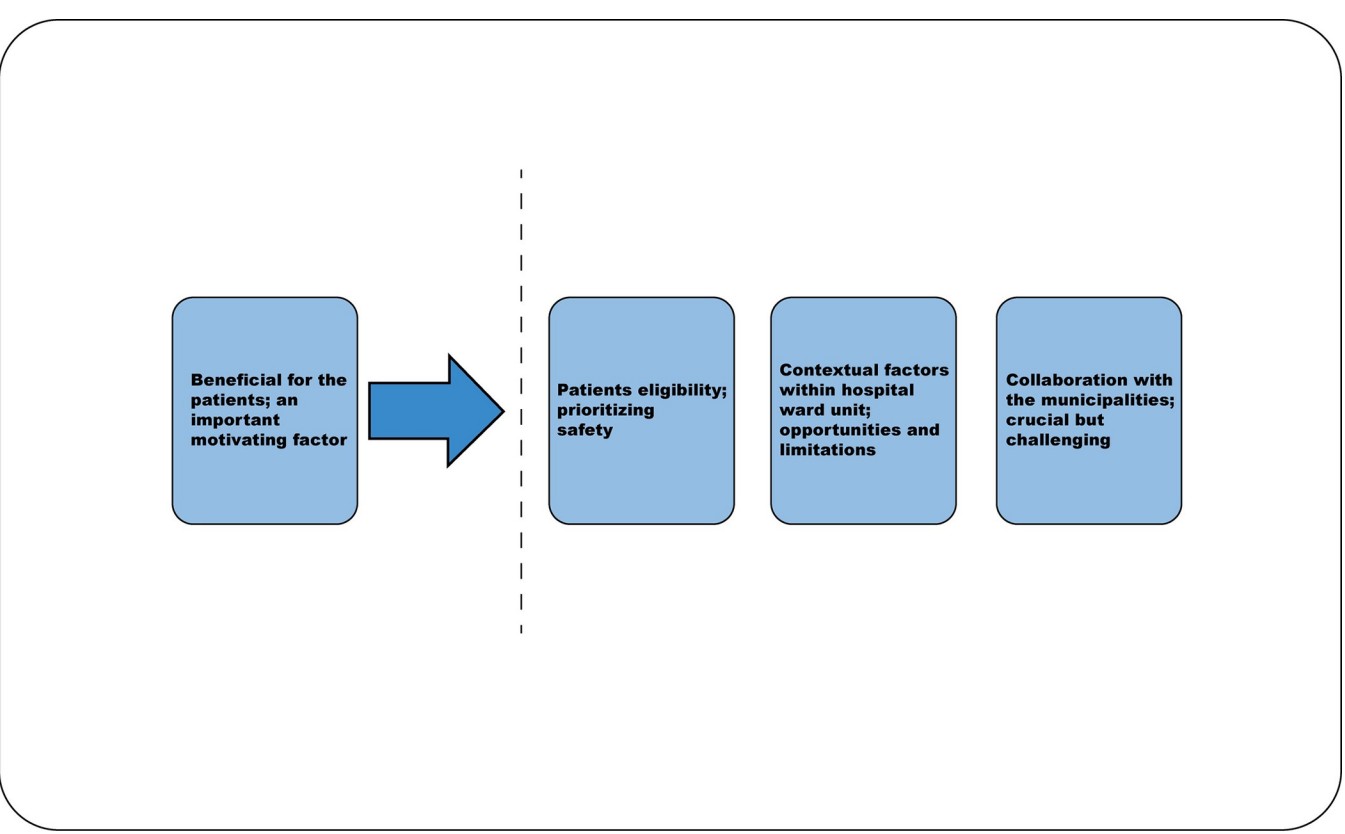

**Fig 1. Main themes of the decision-making process concerning patient enrolment in HaH.**

Furthermore, participants associated HaH treatment with a person-centred care approach. They observed that patients were more actively involved in decision-making and care planning, leading to more tailored treatment and follow-up considering the patients' own premises.

While numerous benefits of HaH were highlighted at the expense of inpatient treatment, staff members preferred hospitalization when patients' overall condition necessitated it or when patients felt unsafe or unwilling to be treated at home for various reasons.

## Patient eligibility; prioritizing safety

Hospital staff considered patient eligibility for HaH treatment based on a set of criteria including diagnostic and clinical aspects, individual patient characteristics, and contextual factors. Hospital physicians, responsible for patient treatment, emphasized the importance of exercising clinical discretion within these predefined criteria.

Patient safety was the overall priority for physicians, who carefully assessed medical risk and patient vulnerability against patient benefits and protective factors. A physician's reflection captured the dilemma posed by a certain patient group:

"Patients whom we describe as multimorbid and who already receive care from home care nurses, you can choose them as well. But the patients should either be so fit that they handle most things themselves or should be followed up very closely. . . so, it is the middle group that is the most difficult to consider" [12].

Safety concerns were closely related to the fear of overlooking exacerbations of patients' conditions or other adverse events after they went home:

"The disadvantage (of HaH) is that you don't get close, so you don't get to monitor them as closely. Don't get to see them daily, that in a way. . . poses a small risk in the sense that you can overlook it if they become septic, or they become ill (. . .)" [13].

Patients living far from the hospital were considered particularly vulnerable. On the other hand, high-quality care in the municipality was seen as a protective factor that could alleviate these concerns.

Another protective factor reported by staff members that could positively influence patient eligibility for HaH was patients'adherence to the treatment plan. Additionally, prior familiarity with patients allowed hospital staff evaluate them more effectively, providing insights into their capabilities, resources, family caregiving support, and social networks at home.

Some staff members perceived safety-related considerations caused physicians to be overly hesitant in identifying patients as eligible for HaH. While acknowledged physicians' dedication to patient safety, they questioned whether the eligibility criteria and safety considerations were overly strict:

"We suggested treating him at home. He was well-functioning, clear, and oriented. But the physicians said *no, no, we must follow up on the infection tests*. We argued that homecare nurses could take daily tests, but the doctors said they needed to see the patient every day" [18].

## Contextual factors within hospital ward units; opportunities and limitations

The decision-making process for patient enrolment in HaH faced challenges due to contextual barriers within the hospital ward units, though enablers were also identified.

In the ward units, several staff members faced challenges integrating new routines and workflows inherent in the decision-making process. These challenges were attributed to a demanding work environment characterized by a high volume of patients, time constraints, and frequent staff turnover:

"You can't really mix the hospital ward with the task of patient enrollment to HaH, which are often chaotic. With too many patients and nurses rushing through the corridors, physicians waiting to operate, everything feels rushed. No one sits down to focus on this task; they are always on their way somewhere. So, I don't think it's a good solution" [3].

Such an environment was incompatible with the time-consuming process of decision-making, which involved thorough observation and extensive data collection to support well-informed decisions. Additionally, the fluctuating conditions of patients necessitated repeated considerations of their eligibility.

Several staff members had limited experience and training in the new HaH routines and tasks, partly because of time constraints and the relatively low number of eligible patients. Physicians with limited experience often sought assistance from more experienced colleagues when considering critical aspects of balancing patient safety with benefits, often leading to delays in the enrolment. However, differences were observed among ward units and staff, with some participants finding the decision-making process less challenging:

"The workflow and logistics worked well because we are used to working in this manner. Those lines have already been drawn, so it's a matter of continuing along the same course. We have established work processes supported by systems, and there haven't been any issues. If we had adopted a completely different approach, we might have experienced challenges" [8].

The work involved in considering patient eligibility for HaH relied on interprofessional collaboration, with hospital staff valuing the complementary skills of all the professionals involved. However, roles and responsibilities were sometimes perceived as unclear. None of the professionals were exclusively dedicated to the tasks associated with HaH, and limited experience and training among staff often led to divergent approaches.

The presence of a few key clinicians in the hospital was highlighted as a positive driver of HaH:

"Enrolling patients to HaH heavily depends on the presence of key individuals in the hospital wards who can make things happen. These persons see no problems, only solutions, and gets things done quickly" [10].

These professionals were highly respected and acknowledged for their positive attitudes towards early discharge services, person-centred care, and home treatment. Some participants perceived that the implementation of HaH relied heavily on these clinicians, expressing scepticism about the service sustainability, as it seemed to fall on the shoulders of a few individuals.

## Collaboration with the municipalities; crucial but challenging

Hospital staff highlighted the importance of a collaboration model between hospital and municipal healthcare services, particularly as such a model offers access to specialized treatment at home to patients residing in sparsely populated areas far from hospitals: "(. . .) that's perhaps the biggest gain we have (from HaH treatment), for those who live furthest away" [12].

Throughout the decision-making process, hospital staff engaged with municipal healthcare professionals to discuss, plan, and provide follow-up care after patients went home. Initially, hospital nurses facilitated contact by calling municipal nurses in the patient's district. Sometimes, insufficient municipal resources, such as staff shortages and lack of necessary expertise, led to the termination of a patient's enrolment in HaH. These instances negatively affected the patient and undermined staff confidence in the HaH program.

Thus, predictability and stability within municipal healthcare services were emphasized, and more binding agreements to ensure comprehensive follow-up at home were encouraged. Despite these challenges, some participants expressed empathy towards the municipal services, reflecting on and questioning the appropriateness of the growing demands placed on them.

When collaborating with municipalities, hospital staff preferred verbal communication and in-person meetings. These dialogues facilitated a common understanding and focus on patient care, which was considered crucial for establishing good relationships with those who would care for the patient at home. As discussed by a physician, time constraints often hindered successful collaboration:

"Some of my best experiences involve meeting face-to-face and reaching agreements instead of writing some e-messages in a hurry to understand patients' needs and important matters. That doesn't always work. When we meet the municipal professional before the

patient goes home, I feel that is better. But it is. . . rare that you get to do something like that" [10].

Hospital staff members shared various experiences with cross-level collaboration, which varied depending on the municipality involved. Challenges often arose from culture and mindset differences between the hospital and municipal healthcare levels. Despite these difficulties, participants acknowledged the potential of the HaH program to improve cooperation across healthcare levels, thus improving these relationships.

## Discussion

The study explores the complex and dynamic decision-making process underlying patient enrolment in HaH. Factors that influenced hospital staff in this process include positive attitudes towards the HaH intervention, patient eligibility, the impact of contextual factors within hospital wards, and the critical yet challenging task of collaborating with municipalities.

Hospital staff expressed positive views and HaH to be beneficial for patients. This perspective was mainly shaped by the perceived drawbacks of hospitalization. How an individual feels about an intervention is a key dimension of the acceptability of complex interventions, influencing healthcare professionals' preferences for one treatment option over others [41]. Hence, it stands to reason that the positive attitudes exhibited by hospital staff motivate patient enrolment in HaH, ultimately enhancing the effectiveness of its implementation [23]. The high acceptability of HaH among hospital staff can be attributed not only to positive feedback from patients and their family caregivers but also to the co-creative service design approach used to develop the current HaH program [34]. Involving all stakeholders, including patients and caregivers, in this process helps identify positive patient outcomes [35]. It is reasonable to assume that this participatory approach could have influenced hospital staff motivation to actively engage with the intervention with the goal of improving patient well-being.

While hospital staff generally acknowledged the benefits of HaH for patients, it became evident that these positive outcomes relied heavily on careful consideration of patient eligibility. Although prior studies have highlighted the safety of HaH [42], concerns regarding patient safety persist. Physicians' reservations regarding potential medical malpractice and patient harm within the HaH setting [25, 43] underscore the strong focus on patient safety, aligning with the fundamental principle of "do no harm" [44]. The safety concerns reported in this study were linked to reduced physical patient monitoring, and the experienced increased risk of overlooking potential occurrence of adverse events and exacerbations. To mitigate these risks and facilitate timely interventions, implementing a system for remote patient observation is recommended [45]. Increasing the use of such technology holds significant promise, not only in enhancing patient safety but also in broadening patient eligibility and improving operational efficiency [46]. However, it is essential to consider several critical factors, including the infrastructure supporting remote technology, the quality and convenience of such technology, and the readiness of healthcare providers and recipients to engage with it. These factors collectively play an important role in the success or failure of the implementation of telemedicine [35, 47, 48].

Our findings may raise the question of whether safety-related considerations were overly strict, potentially limiting the admission of patients who could benefit from HaH treatment. This query captures the complexity of balancing patient safety with ensuring equal access to high-qualitative, person-centred care services such as HaH. Person-centred care is closely linked to the safety of health care delivery [49]. Although these goals often complement each other, conflicts between them can arise [28, 50]. In our study, participants described a complex

balancing act that involved weighing patient risk against benefits, including the expected increases in emotional, social, and psychological well-being. Notably, physicians who had little experience and training sometimes hesitated, sought input, or transferred decisions to more experienced colleagues in situations of uncertainty. This approach could lead to conformity, thus hindering the benefits of diverse perspectives [28]. However, this risk was likely mitigated by the interprofessional approach involved in the HaH decision-making process. Participants valued this approach because complementary skills and knowledge facilitated nuanced discussions and considerations of patient eligibility for HaH [51]. The emphasis on safety should be interpreted in the context of the initial phase of a HaH program, where a cautious approach is appropriate, leading to admission of patients with lower risk profiles [24]. As indicated by Dismore et al. [26], early concerns among clinicians are likely to decrease when they experience the successful delivery of HaH care. The careful initiation of HaH services facilitate a thorough initial assessment of its quality and safety, thus facilitating iterative development [24].

A main environmental challenge faced by hospital staff pertained to the limited time available for patient eligibility considerations. This time constraint is not in harmony with the dynamic and time-consuming nature of the decision-making process, driven by extensive and repeated observations and data collection. Complex interventions that disrupt established workflows and roles are also known to contribute to delays in implementation processes [23]. Research on clinical decision-making has highlighted this issue, revealing that time constraints often force healthcare professionals to prioritize adequate interventions over optimal ones [28]. While guidelines and eligibility criteria can assist hospital staff in the task of streamlining the patient selection process [35], our findings indicate that time-related barriers still remain a challenge. This situation raises the question of whether the automation of support systems could offer a solution that could enhance efficiency and precision in patient selection, potentially alleviating some of the time pressures faced by hospital staff [45].

Hospital staff found it to be less complicated to select patients who were adherent to treatment plans. Treatment adherence is related to high levels of health literacy [52], indicating patients'ability to understand and use information to make decisions and take actions aimed at maintaining their health [53]. Our findings are in line with those of Chua et al. [10], who identified positive health behaviour as a patient characteristic that is suitable for HaH. Despite the advantages for such patients, concerns regarding equity arise [54]. Low health literacy is associated with poorer health and increased health disparities, thus necessitating ethical consideration and effective health literacy interventions to enhance treatment adherence among patients with lower health literacy [53]. Shared decision-making should be fully employed, as patients who are actively involved in choosing their treatment regimens are more likely to adhere to them [55].

Participants embraced the integrated care approach and emphasized the importance of collaborative partnership with municipal healthcare services to broaden the access to HaH care. Unlike typical HaH approaches, which are primarily driven by specialized healthcare services and have limited interaction with community-based services [20, 56, 57], this study emphasizes the HaH programs potential to bridge the gap between hospital-level care and community-based services. This approach broadens access to such care for patients residing in remote areas. Nevertheless, such collaborative models face distinctive challenges [57], as supported by the findings of this study. These concerns primarily focused on the availability of healthcare professionals in sparsely populated areas [45]. This shortage may sometimes hinder municipalities from effectively following up patients in HaH, leading to the exclusion of certain patients from the benefits of HaH treatment. This situation, in turn, may negatively influence the overall acceptance and motivation of HaH among hospital staff, which is crucial to the enrolment of patients in HaH. To mitigate this growing shortage of healthcare professionals, particularly

in primary care, the Norwegian Health Government has proposed to refine the levels of health-care and the roles of professionals as to promote sustainability [58, 59].

## Strengths and limitations

The study had several strengths, including a sample size guided by continuous evaluation of the sample's information power [38] and a rich dataset from the interviews. The high quality of the interviews and the diverse participation of staff members across various professions, levels of experience, ages, and genders enabled a comprehensive exploration of the decision-making process for enrolling patients in HaH. Additionally, the research group possessed diverse backgrounds, including two nurses, one GP, and one neuroscientist focusing on innovation and implementation. This diversity enriched our perspectives on and interpretations of the data.

However, the limitations of this research should be acknowledged. Most participants were interviewed during their working day at the hospital, which may have affected their stress levels and engagement during interviews. While this approach was both practical and desirable for participants in the current study, future research could explore the possibility of conducting interviews outside of regular working hours to minimize stress. To mitigate the risk of participants withholding negative views, the interviewer encouraged the expression of both negative and positive perspectives, emphasizing confidentiality.

Several contextual issues should be considered. The study took place during the initial phase of the HaH service operation at sites without prior HaH experience. Therefore, the transferability of the findings to later phases of service operation should be approached with caution, especially for organizations new to HaH services. Future investigations should examine how decision-making processes evolve over time as the HaH service mature. For instance, longitudinal studies could provide insight to the sustainability of such care programs. Additionally, when considering the potential to apply this analysis to other settings, the integrated care model involving collaboration between hospital and municipal health care should be carefully considered.

## Clinical implications

Healthcare policymakers, managers, and professionals should prioritize overcoming the identified barriers and enhancing corresponding enablers pertaining to enrolling patients in integrated HaH care models. This focus is crucial for both enhancing patient outcomes and optimizing the healthcare system. Active engagement from healthcare professionals in this process is essential for advancing HaH and sharing successful patient experiences. Considerations should be given to implementing remote monitoring technology, particularly in less densely populated areas, to address patient safety concerns effectively. Moreover, exploring the automation of decision-making processes could ease the implementation of HaH in demanding work settings. Promoting health literacy interventions and encouraging shared decision-making with patients are vital for improving treatment adherence. Finally, prioritizing training programs can empower healthcare professionals to make informed, independent decisions, ultimately enhancing the overall effectiveness of HaH.

## Conclusions

Hospital staff provided insights into the complex and dynamic decision-making process involved in considering patient eligibility for HaH enrolment. The findings underscore various barriers and enablers affecting this process, underscoring the necessity to support hospital staff in navigating this complex situation. A key finding is the critical importance of high-quality

decision-making in ensuring positive outcomes and the overall effectiveness of HaH services. Additionally, this study proposes a deeper exploration of the ethical considerations associated with balancing the goal of patient safety with that of equitable access to high-quality, person-centred care within the context of HaH.

## Supporting information

**S1 Checklist. COREQ checklist for qualitative research.**
(PDF)

## Acknowledgments

The authors are grateful to hospital staff members who shared their experiences in this study and to the head of the department and the healthcare professionals who helped recruit the informants.

## Author Contributions

**Conceptualization:** Lillian Karlsen, Bente Prytz Mjølstad, Bjarte Bye Løfaldli, Anne-Sofie Helvik.

**Data curation:** Lillian Karlsen, Bjarte Bye Løfaldli, Anne-Sofie Helvik.

**Funding acquisition:** Lillian Karlsen, Bjarte Bye Løfaldli, Anne-Sofie Helvik.

**Investigation:** Lillian Karlsen, Bente Prytz Mjølstad, Bjarte Bye Løfaldli, Anne-Sofie Helvik.

**Methodology:** Lillian Karlsen, Bente Prytz Mjølstad, Bjarte Bye Løfaldli, Anne-Sofie Helvik.

**Project administration:** Bjarte Bye Løfaldli, Anne-Sofie Helvik.

**Supervision:** Anne-Sofie Helvik.

**Validation:** Lillian Karlsen, Bente Prytz Mjølstad, Bjarte Bye Løfaldli, Anne-Sofie Helvik.

**Visualization:** Lillian Karlsen.

**Writing – original draft:** Lillian Karlsen.

**Writing – review & editing:** Lillian Karlsen, Bente Prytz Mjølstad, Bjarte Bye Løfaldli, Anne-Sofie Helvik.

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
