## [Decision Letter · Decision Letter 0]

8 Aug 2024

PONE-D-24-22161The experiences of hospital staff with decision-making concerning patient enrolment in hospital at home services: a complex and dynamic processPLOS ONE

Dear Dr. Karlsen,

Thank you for submitting your manuscript to PLOS ONE. After careful consideration, we feel that it has merit but does not fully meet PLOS ONE’s publication criteria as it currently stands. Therefore, we invite you to submit a revised version of the manuscript that addresses the points raised during the review process.

**Please revise accordingly**==============================

We look forward to receiving your revised manuscript.

Kind regards,

Zhengmao Li

Academic Editor

PLOS ONE

Journal Requirements:

2. In the online submission form, you indicated that the datasets generated and analyzed during the current study are not publicly available due to the Norwegian law regarding confidentiality and privacy of participants. If desired, selected, de-identified quotes are available from the corresponding author (LK) on reasonable request (lillian.karlsen@helseinnovasjonssenteret.no) 

Reviewers' comments:

Reviewer's Responses to Questions

**Comments to the Author**

1. Is the manuscript technically sound, and do the data support the conclusions?

Reviewer #1: Yes

Reviewer #2: Yes

2. Has the statistical analysis been performed appropriately and rigorously? 

Reviewer #1: Yes

Reviewer #2: Yes

3. Have the authors made all data underlying the findings in their manuscript fully available?

Reviewer #1: Yes

Reviewer #2: Yes

4. Is the manuscript presented in an intelligible fashion and written in standard English?

Reviewer #1: Yes

Reviewer #2: Yes

5. Review Comments to the Author

Reviewer #1: Assessment of the paper

Karlsen L et al. The experiences of hospital staff with decision-making concerning patient enrolment in hospital at home services: a complex and dynamic process.

The paper centres around hospital at home services and some of the barriers to implementation. One such barrier is patient enrolment.

Data collection took place in the form of 22 semi-structured interviews.

The analysis method was reflexive thematic analysis.

Four themes were identified: “beneficial for the patients; an important motivating factor”, “patient eligibility; prioritizing safety”, “contextual factors within hospital ward units; opportunities and limitations”, and “collaboration with municipalities; crucial but challenging”.

One key finding made by the authors is the need to provide support to hospital staff as they navigate the complex situation.

General comments:

This is an interesting paper with information that future administrators and employees working with hospital-at-home concepts may find useful.

To ensure full openness, the authors have ensured the possibility that relevant parts of the transcribed interviews are available (within the Norwegian legislation).

Specific comments:

Scientific principles

The scientific principles applied seem sound. Conducting explorative interview studies from a phenomenological perspective is an approved scientific method in qualitative research. COREQ guidelines were adhered to, increasing the scientific value of the paper.

The sampling of respondents was adequate. The sample size was guided by a continuous assessment of information power. This strengthens the results.

The authors´ qualifications and pre-understandings are described and they were definitely sufficiently qualified to draw their conclusions.

As the guide and interviews were developed for this study and have not previously been published elsewhere, one question needs to be answered: Was the interview guide validated, and if not, how did the authors ensure that the semi-structured guide was comprehensive?

Ethical considerations

The Regional Committee for Medical Research Ethics Central Norway has been introduced to the study and determined that the study did not require ethical approval. This aspect is thus also covered.

Information concerning the participants´ informed consent and right to withdraw is also presented in the study.

Strengths and limitations

The authors carefully balance strengths and limitations, openly addressing the potential limitations that the respondents´ stress levels and engagement may have caused. Furthermore, the interviews were carried out in the initial phases of the establishing HaH, potentially limiting the validity concerning on-to-one transfer of findings to other HaH services.

References

Reference no 1 contains no link. Could the authors mean this link:

https://www.oecd-ilibrary.org/social-issues-migration-health/health-at-a-glance-2021_ae3016b9-en

Reference no 6, a Cochrane review is cited so that it is accessible. However, more thoroughness has been exercised by the authors citing reference no. 14 is also a Cochrane review and is cited correctly with all the necessary information required to gain access to it. The authors should consider the same thoroughness citing reference no. 6 as they do regarding reference no 14.

Reference no. 8 is an unaccepted preprint, available at: https://doi.org/10.21203/rs.2.13538/v1 It is posted in 2019, which is five years ago. This reviewer considers it doubtful to refer to a paper that has not managed to gain acceptance at any journals five years after the initial posting on-line. The authors should seriously consider whether this reference is valid.

Reference no 17. Web-link not present

Ref 44. Web-link not present

Reference no. 60: Is this what the authors would have inserted, enabling the reader to access the reference:

https://www.regjeringen.no/no/dokumenter/stmeld-nr-47-2008-2009-/id567201/?ch=1

Reviewer #2: The paper enetiltled "The experiences of hospital staff with decision-making concerning patient enrolment in

2 hospital at home services: a complex and dynamic process" proposed a decision making analytics for patiaent enrolment in hospital at home services. This paper, from my side, is contributive enough. here are my comments

1. While the analysis is in-depth, I would highly recommend more figures to make your writing more solid.

2. I would recommend a more concised table.

3. I would recommend a more standard formatting for this manuscript

4. I would encourage some ideas regarding how to address those limitations in your future works

6. PLOS authors have the option to publish the peer review history of their article (what does this mean?). If published, this will include your full peer review and any attached files.

Reviewer #1: No

Reviewer #2: No

---

## [Author Response · Author response to Decision Letter 0]

19 Aug 2024

Thank you for the comments and suggestions. We have responded to all comments:

Comments from Reviewer#1 

General comments:

This is an interesting paper with information that future administrators and employees working with hospital-at-home concepts may find useful.

To ensure full openness, the authors have ensured the possibility that relevant parts of the transcribed interviews are available (within the Norwegian legislation).

RE: We are delighted that you find our manuscript interesting for the public.

Scientific principles:

The scientific principles applied seem sound. Conducting explorative interview studies from a phenomenological perspective is an approved scientific method in qualitative research. COREQ guidelines were adhered to, increasing the scientific value of the paper.

The sampling of respondents was adequate. The sample size was guided by a continuous assessment of information power. This strengthens the results.

The authors´ qualifications and pre-understandings are described and they were definitely sufficiently qualified to draw their conclusions.

RE: Thank you for your encouraging feedback.

Scientific principles:

As the guide and interviews were developed for this study and have not previously been published elsewhere, one question needs to be answered: Was the interview guide validated, and if not, how did the authors ensure that the semi-structured guide was comprehensive?

RE: Thank you for pointing out that this aspect was not thoroughly reported. We have now included additional details in the manuscript. Specifically, peers reviewed the interview guide and provided feedback on the questions. Also, a pilot test of the guide was conducted before starting the full set of interviews. The revised text in the manuscript reads: 

“Peers reviewed the questions and provided feedback. A pilot test of the guide was also conducted before the interviews” (lines 159-160).

Ethical considerations:

The Regional Committee for Medical Research Ethics Central Norway has been introduced to the study and determined that the study did not require ethical approval. This aspect is thus also covered. Information concerning the participants´ informed consent and right to withdraw is also presented in the study.

RE: Thank you for your comment.

Strengths and limitations:

The authors carefully balance strengths and limitations, openly addressing the potential limitations that the respondents´ stress levels and engagement may have caused. Furthermore, the interviews were carried out in the initial phases of the establishing HaH, potentially limiting the validity concerning on-to-one transfer of findings to other HaH services.

RE: We are grateful for your comment.

References:

Reference no 1 contains no link. Could the authors mean this link:

https://www.oecd-ilibrary.org/social-issues-migration-health/health-at-a-glance-2021_ae3016b9-en

RE: We are sorry for having omitted the link. The reference is updated in the reference list accordingly. 

References:

Reference no 6, a Cochrane review is cited so that it is accessible. However, more thoroughness has been exercised by the authors citing reference no. 14 is also a Cochrane review and is cited correctly with all the necessary information required to gain access to it. The authors should consider the same thoroughness citing reference no. 6 as they do regarding reference no 14.

RE: Thank you for making us aware of the differences in citing between reference no 6 and 14. We have updated the reference no 6 in the reference list accordingly.

References: 

Reference no. 8 is an unaccepted preprint, available at: https://doi.org/10.21203/rs.2.13538/v1 It is posted in 2019, which is five years ago. This reviewer considers it doubtful to refer to a paper that has not managed to gain acceptance at any journals five years after the initial posting on-line. The authors should seriously consider whether this reference is valid.

RE: We fully agree with the reviewer and have removed this reference and replaced it with: Gonçalves-Bradley DC, Iliffe S, Doll HA, Broad J, Gladman J, Langhorne P, et al. Early discharge hospital at home. Cochrane Database of Syst Rev. 2017.

References:

Reference no 17. Web-link not present

Ref 44. Web-link not present

Reference no. 60: Is this what the authors would have inserted, enabling the reader to access the reference:

https://www.regjeringen.no/no/dokumenter/stmeld-nr-47-2008-2009-/id567201/?ch=1

RE: We have added web-links to references no. 17, 44, and 60 accordingly. Thank you for pointing this out. 

Comments from Reviewer#2

While the analysis is in-depth, I would highly recommend more figures to make your writing more solid.

RE: We have added a figure illustrating the main themes (line 213). Accordingly, we have added text to give a brief information about this illustration (lines 211-212)

I would recommend a more concise table.

RE: We have revised Table 1 (line 144), Table 2 (line 153), Table 3 (line 171), and Table 4 (187), to make them more concise. In our opinion, these revisions are the most effective changes we can make whiteout losing any crucial information for the reader. 

I would recommend a more standard formatting for this manuscript

RE: In line with the formatting guide, we have made following corrections:

- Changed the heading “Background” to “Introduction” (line 60)

- Changed the heading “Methods” to “Materials and Methods” (line 112)

- Reduced the numbers of subheadings under the Materials and Methods section from 7 to 4. The new headings are: 

“Design and setting” (line 113)

“Sample and data collection” (line 145)

“Analysis” (line 172)

“Ethics and reflexivity” (line 188)

- The heading “Discussions” is corrected to “Discussion” (line 352)

- The heading “Conclusion” is corrected to “Conclusions” (line 484)

I would encourage some ideas regarding how to address those limitations in your future works

RE: Thank you for highlighting this, which has helped us improve the quality of our manuscript. We have made the following revisions:

- Added text to the limitation section: “While this approach both was both practical and desirable by participants in the current study, future research could explore the possibility of conducting interviews outside of regular working hours to minimize stress” (lines 457-459).

- Removed and reformulated a sentence from the conclusions (492-494), relocating it to the limitation section “Future investigations should examine how decision-making processes evolve over time as the HaH service mature. For instance, longitudinal studies could provide insight to the sustainability of such care programs” (465-468).

Additionally, we have removed the last sentence from the conclusions section because the suggestion it contained has already been addressed in a previously published paper. Therefore, it is no longer relevant. The sentence we removed was: “Also, we recommend that future studies examine HaH from the perspective of other crucial stakeholders, including primary healthcare providers and patients” (lines 494-496).

---

## [Decision Letter · Decision Letter 1]

8 Sep 2024

The experiences of hospital staff with decision-making concerning patient enrolment in hospital at home services: a complex and dynamic process

PONE-D-24-22161R1

Dear Dr. Karlsen,

We’re pleased to inform you that your manuscript has been judged scientifically suitable for publication and will be formally accepted for publication once it meets all outstanding technical requirements.

Kind regards,

Zhengmao Li

Academic Editor

PLOS ONE

Additional Editor Comments (optional):

Reviewers' comments:

Reviewer's Responses to Questions

**Comments to the Author**

1. If the authors have adequately addressed your comments raised in a previous round of review and you feel that this manuscript is now acceptable for publication, you may indicate that here to bypass the “Comments to the Author” section, enter your conflict of interest statement in the “Confidential to Editor” section, and submit your "Accept" recommendation.

Reviewer #1: All comments have been addressed

Reviewer #2: All comments have been addressed

2. Is the manuscript technically sound, and do the data support the conclusions?

Reviewer #1: Yes

Reviewer #2: Yes

3. Has the statistical analysis been performed appropriately and rigorously? 

Reviewer #1: Yes

Reviewer #2: Yes

4. Have the authors made all data underlying the findings in their manuscript fully available?

Reviewer #1: Yes

Reviewer #2: Yes

5. Is the manuscript presented in an intelligible fashion and written in standard English?

Reviewer #1: Yes

Reviewer #2: Yes

6. Review Comments to the Author

Reviewer #1: My concerns have been fully addressed. I consider this manuscript ready for publication. The manuscript is relevant to administrators considering the hospital-at-home concept.

Reviewer #2: After reviewing the document, it can be seen that the author has addressed all the modification suggestions mentioned

7. PLOS authors have the option to publish the peer review history of their article (what does this mean?). If published, this will include your full peer review and any attached files.

Reviewer #1: **Yes: **Søren Mikkelsen

Reviewer #2: No

---

## [Editor Report · Acceptance letter]

17 Sep 2024

PONE-D-24-22161R1 

PLOS ONE

Dear Dr. Karlsen, 

I'm pleased to inform you that your manuscript has been deemed suitable for publication in PLOS ONE. Congratulations! Your manuscript is now being handed over to our production team.

Kind regards, 

on behalf of

Dr Zhengmao Li 

Academic Editor

PLOS ONE